

# Major biotic stresses affecting maize production in Kenya and their implications for food security

Faith Njeru[1,2], Angeline Wambua[3], Edward Muge[4], Geert Haesaert[5], Jan Gettemans[6] and Gerald Misinzo[1]

[1] SACIDS Africa Centre of Excellence for Infectious Diseases, SACIDS Foundation for One Health, Sokoine University of Agriculture, Morogoro, Tanzania
[2] Department of Veterinary Microbiology, Parasitology and Biotechnology, College of Veterinary Medicines and Biomedical Sciences, Sokoine University of Agriculture, Morogoro, Tanzania
[3] Department of Physical Sciences, Chuka University, Chuka, Kenya
[4] Department of Biochemistry, University of Nairobi, Nairobi, Kenya
[5] Department of Plants and Crops, Ghent University, Ghent, Belgium
[6] Department of Biomolecular Medicine, Ghent University, Ghent, Belgium

Corresponding author
Faith Njeru,
wanjikunjeru2012@gmail.com

## ABSTRACT

Maize (*Zea mays* L.) is a staple food for many households in sub-Saharan Africa (SSA) and also contributes to the gross domestic product (GDP). However, the maize yields reported in most SSA countries are very low and this is mainly attributed to biotic and abiotic stresses. These stresses have been exacerbated by climate change which has led to long periods of drought or heavy flooding and the emergence of new biotic stresses. Few reports exist which compile the biotic stresses affecting maize production in SSA. Here, five major biotic stresses of maize in Kenya are presented which are attributed to high yield losses. They include Maize lethal necrosis, fall armyworm, gray leaf spot, turcicum leaf blight and desert locusts. Maize lethal necrosis and fall armyworm are new biotic stresses to the Kenyan maize farmer while gray leaf spot, and turcicum leaf blight are endemic to the region. The invasion by the desert locusts is speculated to be caused by climate change. The biotic stresses cause a reduction in maize yield of 30–100% threatening food security. Therefore, this review focuses on the cause, control measures employed to control these diseases and future prospective. There should be deliberate efforts from the government and researchers to control biotic stresses affecting maize yields as the effect of these stresses is being exacerbated by the changing climate.

# INTRODUCTION

Maize (*Zea mays* L.) is an important cereal crop in Sub-Saharan Africa (SSA) critical for food security as well as a source of income for millions of small-holder farmers (*Prasanna et al., 2020*). The importance of maize is evidenced by the fact that 90% of the Kenyan population depends on maize for food (annual maize consumption of 4.3 million tons reported in 2018), income, and employment (*Indexmundi, 2018*; *Kusia, 2018*). The total

**Table 1 Average maize production in Kenya (2010–2018).**

| | Maize production | |
| --- | --- | --- |
| Year | Area harvested (ha) | Yield (t/ha) |
| 2010 | 2,008,346 | 1.73 |
| 2011 | 2,131,887 | 1.58 |
| 2012 | 2,159,322 | 1.74 |
| 2013 | 2,123,138 | 1.69 |
| 2014 | 2,116,141 | 1.66 |
| 2015 | 2,098,240 | 1.82 |
| 2016 | 2,337,586 | 1.43 |
| 2017 | 2,086,178 | 1.53 |
| 2018 | 2,273,283 | 1.77 |

maize yield in Kenya ranged from 1.43 to 1.82 t/ha from 2010 to 2018 (FAO, 2018) (Table 1). Similar results have also been reported from data collected from farmers in Kenya who reported average maize yields of 1.48 t/ha for the 2020/2021 cropping season (Njeru et al., 2022).

Maize is cultivated under diversified climatic and agroecological zones mostly by resource-poor farmers with limited access to inputs such as fertilizers and pesticides (Gudero Mengesha, Kedir Mohammed & Sultan Salo, 2019). Despite the low maize production, potential exists for increasing maize yield to over 6 t/ha through increased use of improved seeds and good crop husbandry (Odendo, De Groote & Odongo, 2001). Therefore, there is need to invest in innovative agricultural practices to increase maize yield to be able to meet the demand.

Maize production is threatened by several factors, where biotic factors are of a significant challenge. Biotic factors such as stem borers, and weeds such as *Striga*, have been shown to contribute up to 30% yield loss in maize production (Kusia, 2018). Pathogens such as *maize dwarf mosaic virus*, *sugarcane mosaic virus*, and *wheat streak mosaic virus* have also been reported to cause significant reductions in maize yields in the affected farms.

Globalization and international trade have increased significantly in recent years, subsequently increasing the spread of transboundary pests and diseases. This has led to the emergence of new diseases such as maize lethal necrosis (MLN) and the introduction of new pests such as fall armyworm (FAW) *Spodoptera frugiperda* in Kenya. These new pests and diseases further contribute to low maize production and puts a strain on the agricultural sector.

The population in Kenya is expected to continue increasing in the coming decades reaching 66 million in 2030; however, food production is not expected to be on pace. Hence, the purpose of this review article is to present the major biotic stresses affecting maize production in Kenya, with a focus on two pests namely: fall armyworm and desert locusts, two fungal pathogens: gray leaf spot (GLS), and Turcicum leaf blight (TLB) and

one viral disease: maize lethal necrosis (MLN). These biotic stresses are very common, occurring every maize growing season and have the potential to cause yield losses of 100%.

This review article targets plant breeders, pathologists, scientists in the seed companies and students in agricultural research. Several previous review articles have been written on different aspects of biotic stresses affecting maize production such as (*Redinbaugh & Stewart, 2018*; *Prasanna et al., 2022*; *Shiferaw et al., 2011*). These previous review articles have focused on a single specific biotic stress. The current review article combines several biotic stresses affecting maize production.

## SURVEY METHODOLOGY

The studies used to write this review article were identified from science direct, research gate and web of science. Only articles written in English were reviewed. The inclusion criteria included original articles and previous review articles on biotic stresses. The search terms used included maize lethal necrosis, fall armyworm, desert locusts, gray leaf spot; turcicum leaf blight. The search terms were used with the Boolean operator "AND" maize production. This enabled to focus the search to articles written in reference to maize cultivation.

### Biotic stresses

Productivity in plants is reduced by biotic stresses caused by living organisms such as pests, bacteria, fungi, weeds and viruses. Economic losses to biotic stresses vary depending on the stress and the affected crop. A study by *Pratt, Constantine & Murphy (2017)* reported that biotic stresses in Kenya cause losses of between 3.8 to 123.6 million USD.

Maize is cultivated in 40% of the total crop area in Kenya and its production is often affected by biotic stresses. These biotic stresses affect maize during growth and also after harvest and they not only impact the yield but also the quality of the crop harvested.

Below we discuss in detail the main biotic stresses of maize which are of national importance in Kenya as they cause significant yield losses disrupting food security.

### Fall armyworm

#### History and biology of FAW

Fall armyworm (FAW) *Spodoptera frugiperda* (Fig. 1) is a type of caterpillar that disperses to long distances annually (*Zaman-allah et al., 2019*). The female moth lays eggs in masses on the leaves of many plants (*De Groote et al., 2020*). After hatching, the larvae, which is the most destructive stage of the pest, disperses throughout the crop field consuming vegetation they come across (*Capinera, 2008*). FAW is a polyphagous pest that attacks over 60 plant species, but its major effects have been on a few crops such as rice, millet, sorghum, and maize (*Sisay et al., 2018*).

FAW is native to the tropical regions of the western hemisphere from the United States to Argentina (*Marenya et al., 2022*). FAW is relatively new in Africa, where its first report was in 2016 in West Africa following distress calls by maize producers of high armyworm populations (*Goergen et al., 2016*). Subsequently, FAW was reported in other African
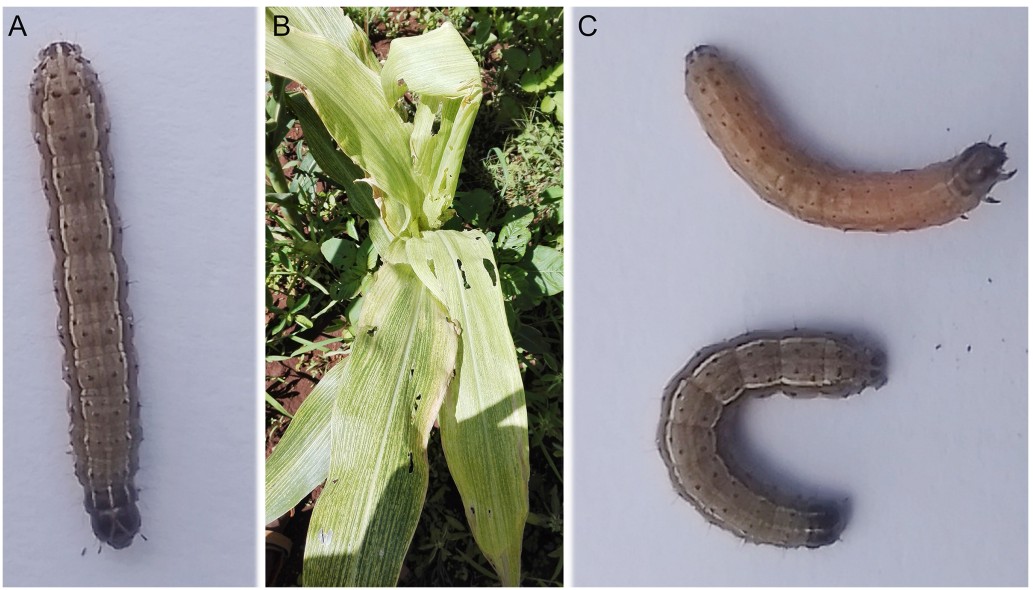

**Figure 1 Fall armyworm.** (A) and (C) is the photo showing the larval form of FAW (taken by Paul Njau at Naivasha, Kenya in 2023). (B) Photo of maize plants affected by FAW in Kenya 2020 (photo taken by Faith Njeru).

countries including Kenya in 2016 (*Mutyambai et al., 2022*) and by 2018, FAW had affected 86% of the maize farmers in Kenya (*De Groote et al., 2020*).

The fast spread of FAW is attributed to its natural distribution capacity (strong migratory pest, producing many eggs) and the increase in international trade (*Tambo et al., 2020*). Additionally, its preference for maize, the major cereal crop in Africa also aids in the fast spread of this pest which has been reported in all Sub-Saharan African countries except Lesotho (*Tambo et al., 2020*).

### Impact of FAW on maize production

Damage on maize by FAW is observed on all plant parts and at all maize developmental stages, with the pest reported to affect the stem base of maize plantlets, the leaves of maize at the vegetative stage, and also on grown maize plants with the pest able to feed on tassels or bore into ears (*Goergen et al., 2016*). It has been identified mostly in the Southeastern U. S. as a regular and serious pest destroying maize crops (*Zaman-allah et al., 2019*) causing yield losses of up to 57% in Latin America, depending on the crop season and the maize variety planted (*Burtet et al., 2017*).

Results based on socioeconomic surveys and farmer estimates of yield loss, undertaken in different countries in Africa have estimated maize yield loss due to fall armyworm to be in the range of 22–67% (*Kansiime et al., 2019*). A community based survey done in Kenya in 2017 and 2018 by *De Groote et al. (2020)* reported average yield losses of 33% to FAW. These losses due to FAW are similar to results reported by a farmer survey in Ghana and Zambia which reported yield losses of 26–40% and 35–40% respectively (*Nboyine et al., 2020*) and 32% in Ethiopia (*Baudron et al., 2019*).

*Kumela et al. (2019)* reported FAW-induced maize yield reductions of 32% (Ethiopia) and 47% (Kenya), based on a survey of maize farmers while maize yield reductions of 27% and 35% were reported in Ghana and Zambia respectively (*Rwomushana et al., 2018*). However, in Zimbabwe, a study by *Baudron et al. (2019)* estimated FAW-induced maize yield reductions of 12% based on a rigorous field scouting method.

In addition, FAW has been shown to significantly increase the vulnerability of maize to other secondary plant pathogens since it destroys the protective husk of maize. The larvae of FAW causes damage to corn ears and the kernels (*Herrington et al., 2014*). The damage caused by FAW to maize kernels has been shown to increase the growth of certain fungi that cause mycotoxins (*Devi, 2018*). Mycotoxins are toxic metabolites with mild to chronic health effects to humans and animals (*Wokorach et al., 2021*).

The presence of this new pest in most SSA countries adds to the threat caused by native lepidopteran maize stalk or ear borers such as African maize stalk borer (*Busseola fusca*). Besides its direct effect on agricultural production, FAW also has the potential to significantly affect access to foreign markets (*Goergen et al., 2016*). Maize is an important crop as a source of food and income to millions of smallholder farmers in SSA, FAW, therefore, poses a significant threat to food security in SSA and the attainment of sustainable development goal 2 to end hunger by 2030.

### FAW detection and control

An assay based on loop-mediated isothermal amplification (LAMP) has been reported for the species specific diagnosis of FAW (*Osabutey et al., 2022*). However, for the small scale farmer, scouting protocols which looks for the signs of egg hatch and feeding by egg larvae instar have been developed to allow early detection of FAW for better management (*Prasanna et al., 2018*). Besides scouting, sex pheromone traps have been developed to track the presence and movement of FAW in a certain region (*Prasanna et al., 2018*).

FAW is new to the African continent, therefore control measures are limited and little is known about the most effective agronomic practices for FAW control (*Baudron et al., 2019*; *De Groote et al., 2020*). Therefore, most farmers are left with the option of using chemical pesticides for FAW control. This option is expensive and poses risks to health and the environment (*Sisay et al., 2018*). A study done across five African countries of Ghana, Rwanda, Uganda, Zambia and Zimbabwe showed that use of pesticides for FAW control was the most preferred management option among the surveyed farmers (*Tambo et al., 2020*).

The common pesticides used by the farmers in Africa include profenofos and cypermethrin traded as rocket and lambda-cyhalothrin (Karate). These are broad spectrum non systemic pesticides having contact and stomach action. Unfortunately, some farmers reported using pesticides such as monocrotophos, dichlorvos, methomyl, and methamidophos which are highly toxic and prohibited products (*Tambo et al., 2020*). Due to the devastating effect of FAW on maize plants and the limited know how on its management, farmers resulted in indiscriminate use of pesticides with some farmers applying fungicides such as mancozeb (*Tambo et al., 2020*).

In addition, the use of pesticides has been reported to be ineffective due to the larval behavior of this pest; larval stage bores into the host plant developing under a protected environment, making it difficult to reach with target insecticidal sprays (*Burtet et al., 2017*). FAW has also been reported to have evolved resistance to several insecticides used including carbamates, organophosphorus, and pyrethroids (*Wan et al., 2021*). FAW have been shown to develop resistance to insecticide by metabolic detoxification mechanism (*Wang et al., 2022*). The increased activity of glutathione S-transferases (GSTs) resulted in FAW resistance to pyrethroids. Another mechanism that has been reported for FAW resistance to insecticides is the target resistance mechanism (*Wan et al., 2021*).

In Brazil, the control of FAW has been through the use of transgenic plants encoding the Cry1F gene from *Bacillus thuringiensis* (Bt) (*Bernardi et al., 2015*). However, some studies done in Brazil have shown Bt maize plants expressing the Cry1 proteins to be affected by FAW as a consequence of field evolved resistance to Cry1 proteins (*Burtet et al., 2017*). Maize plants expressing the Vip3A gene product from Bt were shown to be effective against FAW due to the high toxicity of Vip3A protein to FAW (*Burtet et al., 2017*).

Though the use of Bt maize expressing the Cry1 and Vip3A proteins have been shown to have significant success in the control of FAW, the use of this technology need to be monitored and integrated with other control options to control the evolution of resistance against the Cry1 and Vip3A proteins by FAW.

However, the ethical concerns for use of genetically modified (GM) plants still restrict the use of such plants where legal restrictions are imposed on GM plants. Therefore, in most SSA countries, more research is being carried out to identify alternative control strategies that are effective, adaptable, and applicable to the smallholder farmer's in SSA.

These mitigation strategies include cultural, the use of bioagents, and the use of resistant plant genotypes as extensively reviewed by *Kasoma et al. (2020)*. Natural enemies of FAW which include parasitoids that are small insects that develop attached to the host and eventually kill the host, have been reported for FAW. *Sisay et al. (2018)* reported several species of parasitoids for FAW in Ethiopia, Kenya and Tanzania with parasitism ranging from 4 to 45.3%. The parasitoids that have been reported for FAW include *Cotesia icipe*, *tachinid fly*, *Palexorista zonata*, *Charops ater* and *Coccygidium luteum* (*Sisay et al., 2018*). It has also been shown that intercropping with plants such as Tephrosia or desmodium which produce repugnant chemicals help to repel the adult female moths, reducing the number of eggs laid on host plants (*Prasanna et al., 2018*).

In addition to the parasitoids which are natural enemies of FAW, entomopathogenic microbials which include fungi, bacteria, viruses and nematodes have also been reported to be effective against FAW through laboratory assays and in the field (*Hruska, 2019*). Entomopathogenic fungi *Beauveria bassiana, Metarhizium rileyi, and Metarhizium anisopliae* have more than 50% mortality rate of FAW eggs and larvae (*Guo et al., 2020*). The nematode *Steinernema feltiae*, when sprayed on maize ears in the field was shown to effective against FAW (*Fallet et al., 2022*).

Microbials have also been shown that when applied in combination can lead to more efficient FAW control. Simultaneous application of *Metarhizium rileyi* strains and nucleopolyhedroviruses (NPV) had a better control on FAW (*Souza et al., 2019*).

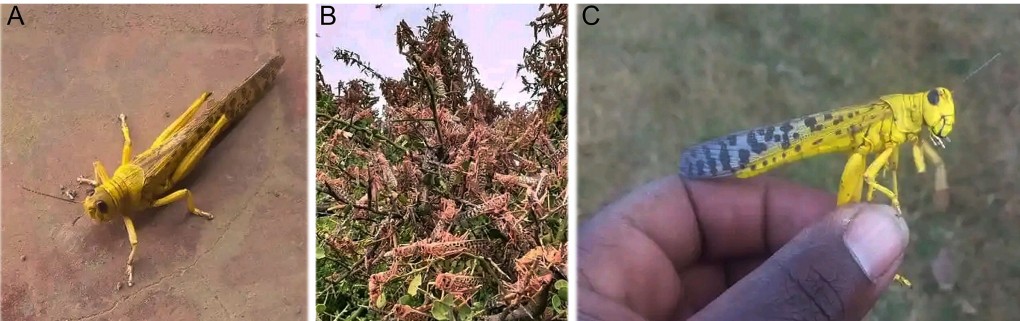

**Figure 2** (A) and (C) are the images of desert locust. (B) Swarm of desert locust in a field in Baringo, Kenya (photo taken by Isaac Ruto).

Microbials have been used to develop biopesticides for FAW control, which are less hazardous to the environment and the farmers. Few biopesticides are registered for use in Africa, therefore more research should be done to encourage the use of biopestides in Africa (*Bateman et al., 2018*).

A study by *Baudron et al. (2019)* in Zimbabwe, reported that FAW damage was frequently reduced by frequent weeding and by minimum and zero tillage. However, pumpkin intercropping was found to significantly increase FAW damage. Further research is needed to determine which crops are the most efficient in controlling FAW and acceptable to farmers. Farmers' have also been shown to employ a range of cultural and physical practices, based on indigenous knowledge *e.g.*, hand picking of egg masses and caterpillars, and application of ash/sand on the larvae, some with considerable levels of success (*Kansiime et al., 2019*).

Some of the cultural practices used by the farmers have been studied and shown to be effective. For example, the use of neem oil was reported to have 70% mortality of FAW larvae (*Matova et al., 2020*). Rabbit urine has also been shown to be effective against FAW by reducing larval feeding (*Kemunto et al., 2022*). Therefore, potential exists for the application of indigenous knowledge in the control of FAW especially in small scale farming.

Some maize varieties were found by *Baudron et al. (2019)* to be tolerant to FAW damage. Therefore, there has been research to identify quantitative trait loci (QTL) for application in marker-assisted selection (MAS) to facilitate the breeding process. *Womack et al. (2020)* identified two important QTL explaining 37% of the phenotypic variance of leaf-feeding damage by FAW in maize. The resistant variety used, Mp705 was responsible for the leaf-feeding damage-reducing alleles for both large-effect QTL and most of the small-effect QTL identified in the study by *Womack et al. (2020).*

There is a need for the development and implementation of evidence based efforts to control this pest. Development of alternative control and preventive methods for FAW, based on agronomic management would be more affordable to resource-constrained small holder farmers with less risk to health and environment. But studies are still ongoing to understand the unknown dynamics of establishment, spread, and environmental conditions favoring the survival of FAW in the continent.

## Desert locusts

### Emergence of desert locusts

The desert locusts, *Schistocerca gregaria* (Fig. 2) are mainly found in arid and semi-arid areas spanning regions from West Africa to India. Desert locusts' outbreak and invasion date back to 1860. Since then, different outbreaks have been recorded over the years some even lasting for over 22 years (*Lecoq, 2003*). The recent outbreak can be traced back to the 2018 cyclones coupled with the warm weather and heavy rainfall experienced at the end of 2019 (*Roussi, 2020*). In this outbreak, Kenya has experienced its worst invasion in 70 years where a large swarm occupied an area of about 2,400 square kilometers (*Roussi, 2020*).

Current outbreaks are worsened by the current state of political instability, under-financed control centers, the COVID-19 pandemic and poor early detection strategies (*Roussi, 2020*). Therefore, with the recent advances in technology and remote sensing, there is need to devise better and effective early sensing tools for arresting the situation before much damage is done.

### Losses caused by desert locusts

At a local level, desert locusts can cause desertification due to the raged soils caused by depletion of vegetation cover. Development of irrigation schemes in SSA have also aggravated the situation by providing a favorable environment for breeding sites of the desert locusts, consequently leading to recurrent outbreaks (*FAO, 1994*). Desert locusts can migrate over very long distances attacking all types of vegetation cover hence posing great threat to agro-sylvo-pastoral production (*Lecoq, 2003*). In times of favorable weather conditions of high rainfall (favorable environment for laying eggs), outbreaks, upsurge and invasions usually occur threatening food security.

On the flip side, insects have over the years been used as an alternative source of high-quality proteins, energy, fats and mineral elements. Together with migratory locusts, desert locusts are consumed for proteins and fats although research on its actual nutritive value has not been reported (*Salama, 2020*; *Mariod, 2020*). There is, therefore, need to investigate the actual nutritive value and the data obtained used to sensitize the society on the importance of such sources of food. Moreover, specific elements of the same can be used in fortification of staple foods like maize and rice in poor communities where malnutrition is rampant.

### Control of desert locusts

Over the years, the control for desert locust outbreaks and invasion has been the use of organo-phosphorous insecticides such as the widely used chlorpyrifos. These compounds are not only harmful to humans and other living organisms but also pollute the environment contributing to the chlorofluorocarbon (CFCs) and global warming at large (*Gillespie, Burnett & Charnley, 2000*). Therefore, researchers are advocating for better monitoring and alternative measures to synthetic pesticides to control these insects.

First in the list of alternatives, are biopesticides. Research had shown that a fungus, *Metarhizium anisopliae* can kill desert locust by growing in the insect's body. *Gillespie, Burnett & Charnley (2000)* and his team reported that these entomopathogenic fungi kill

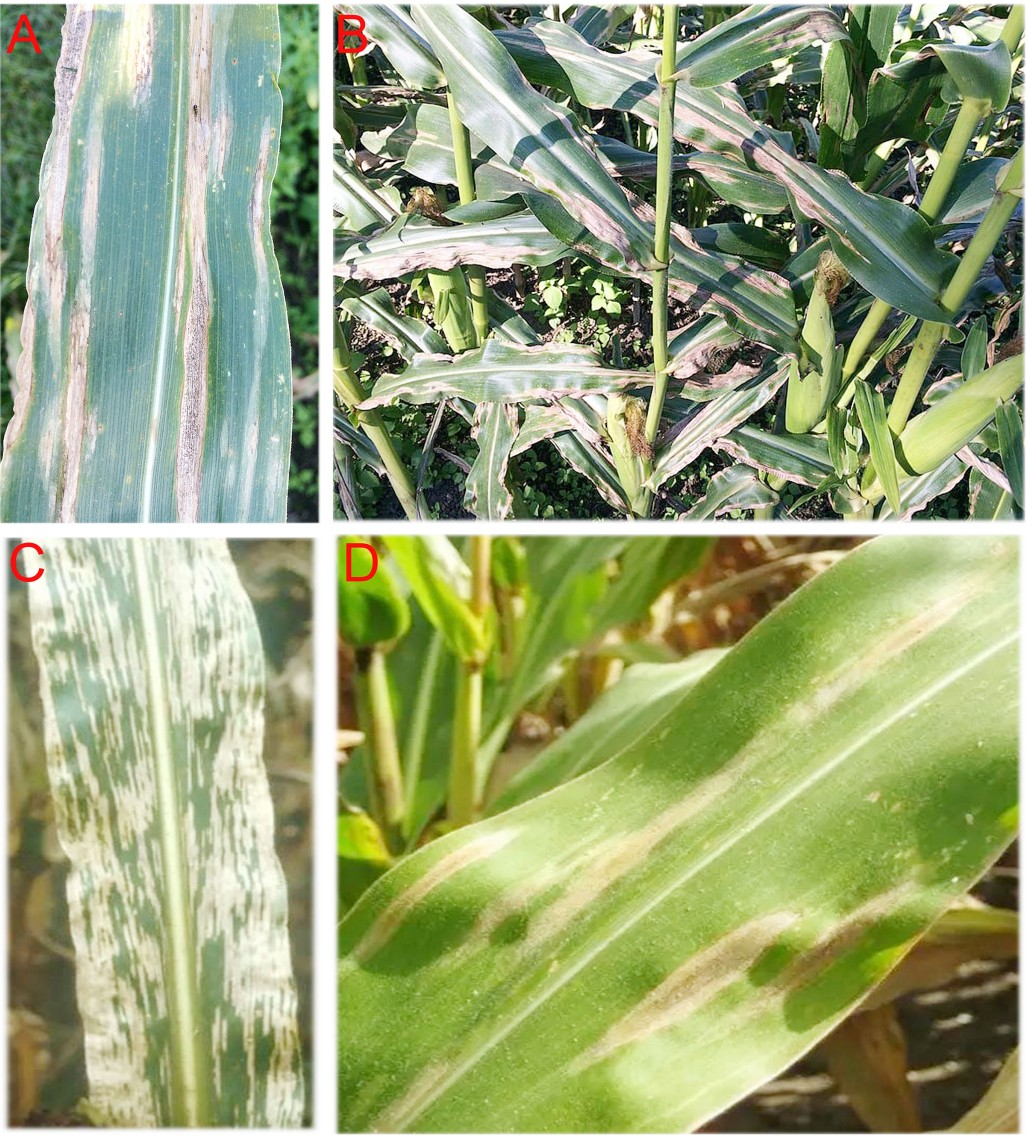

**Figure 3 Photo of maize plants showing symptoms of Turcicum leaf blight (A) and (B) and Grey leaf spot, (C) and (D).** Photo taken by Paul Njau at Naivasha, Kenya.

the desert locust around day four and five of infection. Therefore, this bio control strategy although it is environmentally friendly, does not offer immediate results but can be used in combination with other control methods. Hence, further research on the appropriate combination of control strategies need to be done to establish the ones that will give optimal results.

Recently, *Galal & Seufi (2020)* characterized the microbial community present in the desert locust's body. Using morphological data and molecular identification techniques, nine bacteria species were identified from the *S. gregari* adult body. The identified bacteria species were *E. faecalis, L. paracasei Bacillus sp., S. epidermidis, Escherichia sp., Salmonella sp., Pluralibacter sp., Shimwellia spp*; while one species was unclassified. However; further

research on the role of the identified microbes on the survival of the locusts can give an insight on whether the said microbes can be used in the control of the locusts. In addition, further studies are needed to establish the inter-relationship between these microbes and the entomopathogenic fungi *Metarhizium anisopliae*.

Another environmental friendly control strategy is the use of crude extracts from Jatropha (*Jatropha curcas L.*) and neem (*Azadirachta indica A. Juss*) which have been demonstrated to be toxic, antifeedant, and growth regulating compounds against nymph stage of the desert locusts (*Bashir & El Shafie, 2014*). However, further investigation and fractionation of the extracts is required to characterize the specific compounds which are effective against desert locusts in these plants. There is also need to establish the effectiveness of these plant extracts on hopper stages of the locusts. Moreover, studies on the effect of combined extracts from both plants on the effect on the nymph and hopper stages of the locusts are yet to be reported.

## Gray leaf spot and turcicum leaf blight
### History of GLS and TLB to maize production
Gray leaf spot (GLS) (Fig. 3) caused by *Cercospora zea-maydis* (*C. zein-maydis*) (*Tehon & Daniels, 1925*), is a maize leaf disease that is a global threat to maize production. Molecular research shows that *C. zea-maydis* is mainly distributed in Brazil and African countries, and is predominant in the United States of America (*Ward et al., 1999*). In Kenya, GLS was first reported in 1993 in the Western region and has since spread to other maize growing areas in the country (*Danson et al., 2008*). Turcicum leaf blight (TLB) (Fig. 3), is distributed widely around the world and is caused by *Helminthosporium turcicum* affecting maize plants from seedling to harvest stages (*Karavina, Mandumbu & Mukaro, 2014*). It is a serious problem in North Eastern United States, sub-Saharan Africa, and areas of China, Latin America, and India (*Adipala, Takan & Ogenga-latigo, 1995*, *Dharanendraswamy, 2003*).

### Impact of GLS and TLB to maize production
Documented yield losses attributed to GLS range from 11% to total yield loss in cases of severe infection (*Liu et al., 2016*). GLS disease is characterized by necrotic lesions leading to leaf senescence hence reducing the photosynthetic capacity of the plant. This reduced photosynthetic capacity is linked to poor grain filling and stalk lodging which ultimately lead to poor maize yield (*Gethi et al., 2013*).

TLB is characterized by elliptical grey-green lesions which turn to tan with dark spots of sporulation as maturation continues and eventually leads to defoliation affecting grain yield as seen in GLS (*Hooda et al., 2017*). Yield losses range from 15% to 70% depending on the onset and severity of the infection (*Hooda et al., 2017*). Wet and humid weather conditions favor the spread and increase the severity of GLS and TLB (*Jakhar et al., 2017*). Crop yield losses threaten global food security especially in SSA where farming is the main source of livelihood.

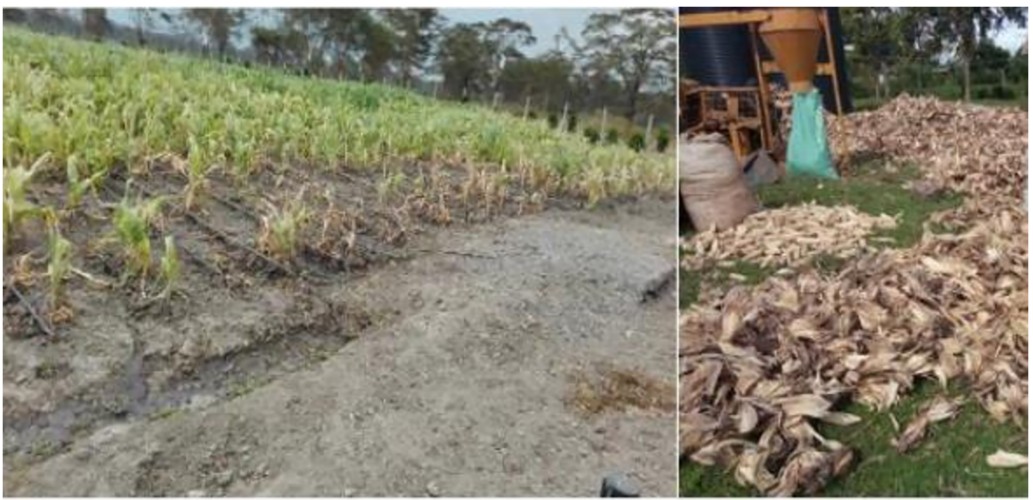

**Figure 4** **Maize lethal necrosis.** (A) MLN infected field under artificial inoculation at CIMMYT-KALRO MLN screening facility. (B) maize harvested from an MLN infected field in Kenya. Photo credit: Faith Njeru.                                                            

### Control strategies for GLS and TLB

The common control methods deployed for these two maize diseases are mainly cultural, chemical and host plant resistance. In the cultural approach, crop rotation and effective burying of the infected plants debris, have reduced the severity of infection in the subsequent round of planting. In addition, effective removal of the over-wintering infected debris, reduced the inoculum available for the next season hence reducing the infection pressure. Fungicides have also been used to control TLB and GLS though it's less cost effective, leads to fungicide resistance and the continued use of chemicals has raised environmental concerns. Due to these drawbacks, cultural and chemical methods have proved to be uneconomical and unreliable especially for the poor small scale farmers.

Host plant resistance therefore, remains the most viable and reliable method for the management of TLB and GLS. Breeders and researchers around the world have hence embarked on research aimed at identifying host resistant varieties for different biotic and abiotic stresses in different economically important crops. For instance, *Karavina, Mandumbu & Mukaro (2014)*, identified three maize hybrid lines (053WH54, ZS225 and SR52) which were resistant to TLB diseases. In addition, GLS resistant genes were established in various researches (*Gethi et al., 2013*; *Kuki et al., 2018*; *Lehmensiek et al., 2001*).

Due to increasing population and climate change, the demand for food is expected to increase. Therefore, there is need to search for more resistant maize lines for both biotic and abiotic stresses and give farmers a wide variety of maize lines. In recent decades, the invention of next-generation sequencing (NGS) technology and high throughput phenotyping platforms have paved ways for fast and effective means of identifying host resistance, molecular breeding and marker assisted selection. In addition, new

biotechnological tools have made it possible to perform gene editing and produce superior breeds.

## Maize lethal necrosis

### History of MLN

Maize lethal necrosis (MLN) (Fig. 4) is a disease of maize that was first reported in Peru in 1973 and then in USA in 1976 (*Hutchens, 1978*). It is a viral disease caused by the synergistic interaction of *Maize Chlorotic Mottle Virus* (MCMV) and any of the viruses belonging to the *potyviridae* family (*Eunice et al., 2021*). The first report on MLN in the African continent was in Kenya in 2012 (*Wangai et al., 2012*). The disease was later reported in several other African countries (*Semagn et al., 2015*) including Uganda, Tanzania, Ethiopia, Rwanda, D.R Congo, and South Sudan (*Eunice et al., 2021*). These countries are known as MLN endemic countries.

*Maize dwarf mosaic virus* (MDMV) and *wheat streak mosaic virus* (WSMV) are the main potyviruses combining with MCMV to cause MLN in the USA (*Uyemoto, Bockelman & Claflin, 1980*) while *sugarcane mosaic virus* (SCMV) has been implicated in MLN infections in Africa (*Mahuku et al., 2015*). Recently, a study by *Stewart et al. (2017)* showed the *Johsongrass mosaic virus* (JGMV) to cause MLN in co-infections with MCMV in East Africa. *Maize yellow mosaic virus* (MaYMV) has been discovered in deep sequencing studies in MLN-infected maize plants. However, it was shown that it does not cause MLN but it significantly enhances MLN symptoms such as stunting (*Stewart & Willie, 2021*).

Maize is the main natural host for MCMV; however, it has also been detected in millet, Johnsongrass, *Digitaria* sp., sedge, *Setaria* sp., and sugarcane (*Mahuku, 2019*). Common SCMV hosts reported in Africa include sugarcane, maize, sorghum, Kikuyu grass as well as other Poaceous plant species.

A study conducted on the metagenomic analysis of viruses causing MLN in Kenya showed MCMV as the most prevalent virus in maize-growing regions in Kenya (*Wamaitha et al., 2018*). SCMV population in Kenya was shown to be diverse, consisting of numerous strains distantly related to isolates from other parts of the world (*Wamaitha et al., 2018*). Limited sequence divergence among MCMV isolates has been reported in population genetics studies (*Braidwood et al., 2018*). Phylogenetic analysis has shown similarity between MCMV isolates found in East Africa to those found in China (*Braidwood et al., 2018*).

### Losses caused by MLN on maize production

Maize plants infected with MLN show more severe symptoms than plants infected with MCMV or SCMV alone (*Gudero Mengesha, Kedir Mohammed & Sultan Salo, 2019*). MLN-infected plants show a wide range of symptoms depending on the maize variety, time of infection, and prevailing environmental conditions (*Miano, 2014*). The symptoms are: chlorotic mottle on leaves, developing from the base of young whorl leaves upward to the leaf tips; mild to severe leaf mottling; and necrosis (*Prasanna et al., 2020*). Necrosis of young leaves leads to a "dead heart" symptom, and plant death (*Wangai et al., 2012*). Severely affected plants have small cobs with little or no grain set (*Wangai et al., 2012*).

Results from a community-based survey done by *De Groote et al. (2016)*, estimated the proportion of maize lost in the community, at 0.5 million tons per year, or 22% of the average annual production before MLN, with an estimated economic loss of $180 million (*Beyene et al., 2017*). The most affected region by MLN in maize production is Western Kenya with more than half of the farmers affected and with a 58% loss of maize (*Beyene et al., 2017*). Central and Eastern Kenya, had up to a third of the farmers affected and with 19% maize yield loss (*Beyene et al., 2017*). Many studies have estimated the losses due to MLN in different maize agro-ecological zones in Kenya to be between 23–100% (*Njeru et al., 2022*).

Different studies have shown than MLN disease can completely wipe out maize plants leading to 100% loss in yield. This is especially devastating to millions of families and small holder farmers who depend on maize as a source of food and income. The effect of MLN is also felt by small and medium enterprises (SME's) seed companies and processors (*Prasanna et al., 2020*). Demand for seed of commercial seed varieties declined when MLN was major epidemic with losses of sales for the companies (*Prasanna et al., 2020*).

Due to the significant impact of MLN on the maize sector, International Maize and Wheat Improvement Center (CIMMYT) in collaboration with National Agricultural Research System (NARS) and National Plant protection Organization (NPPO) have put in place several mechanisms to prevent its further spread to the MLN non endemic countries where MLN/MCMV has not yet been reported (*Prasanna et al., 2020*). The mechanisms that have been put in place include diagnosis of MLN-causing viruses in maize seed, monitoring and surveillance of MLN across Africa, production and exchange of MLN pathogen free commercial maize seed (*Prasanna et al., 2020*).

### MLN epidemiology

Sustainable control of plant diseases requires an understanding of the disease including the biology of the pathogens causing the disease, suitable environmental factors favoring host-pathogen interactions and vectors or any other means involved in the spread of the disease. Epidemiological studies on the spread of MLN have identified the spread of the disease from plant to plant and from field to field to be mainly through insect vectors (*Gudero Mengesha, Kedir Mohammed & Sultan Salo, 2019*).

Several insect vectors such as thrips, maize rootworms, leaf beetles and leaf hoppers have been associated with the spread of MCMV (*Gudero Mengesha, Kedir Mohammed & Sultan Salo, 2019*). Aphids on the other hand are the main insect vectors associated with the spread of SCMV (*Brault et al., 2010*). Seed transmission of MCMV has been reported though at very low rates (0.03–0.33%). This may partly explain how MCMV has managed to travel across continents and countries. MLN has also been shown to be in the soil, irrigation water, and infected plant debris (*Kinyungu et al., 2019*; *Miano, 2014*).

### Disease diagnostics

To detect MLN, you have to be able to detect both MCMV and SCMV (causative viruses for MLN). Commercial enzyme linked immunosorbent assay (ELISA) kits based on polyclonal antibodies are available for MCMV and SCMV detection. Monoclonal

antibodies have also been developed and shown to have sensitive and specific detection of MCMV (*Wu et al., 2013*). Gene amplification techniques including reverse-transcriptase polymerase chain reaction (RT-PCR) (*Awata et al., 2019*) and reverse transcriptase loop mediated isothermal amplification (RT-LAMP) (*Mwatuni et al., 2020*) have been developed for MCMV and SCMV detection.

Use of symptoms to tell the presence of MLN is challenging because symptoms caused by MLN viruses vary depending on the age of maize plant and environmental conditions. In addition, symptomless plants have been found to be MCMV positive. Therefore, diagnostic methods are significant to validate the presence of MCMV and SCMV.

### Control of MLN

A study conducted by CIMMYT in collaboration with Kenya Agricultural and Livestock Research Organization (KALRO) between 2012 and 2014 found that, most of the maize germplasm had low levels of resistance to MLN (*Semagn et al., 2015*). Therefore, more than 95% of the inbred lines and hybrids grown by farmers in Kenya are susceptible to MLN (*Semagn et al., 2015*). With the continuous cultivation of maize throughout the year, this exacerbates the problem caused by MLN. Therefore, more research is needed to generate and release resistant hybrids to MLN.

Breeding for resistance following the conventional method is time consuming and costly. Therefore, many researchers are applying genomic selection as a promising breeding tool to improve the efficiency and speed of the breeding process. MCMV resistance has been shown to occur in different chromosomal locations and five QTL for MCMV resistance have been found in chromosomes 3, 5, 6 and 10 (*Jones et al., 2018*).

Both major QTL effects and multiple minor effects QTL for MCMV resistance were identified (*Jones et al., 2018*). Similar observations were made in a subsequent study by *Awata et al. (2020)* that identified QTL for resistance to MCMV in seven bi-parental populations where some QTL showed major effects in some populations and minor effects in other populations.

To further understand the resistance mechanism of MCMV, *Jones et al. (2018),* tested systemic leaf tissue from five inbred lines for the presence of MCMV and determined that ELISA responses for tissue of inoculated plants from the five lines and the susceptible control (Oh28) were similar. Therefore, only tolerance to MCMV have been identified rather than resistance. The responses of MCMV and SCMV viruses are not linked as the SCMV-resistant line Pa405 had no resistance to MCMV (*Jones et al., 2018*).

MLN is a complex disease with multiple reservoirs and transmission pathways. There is need for continued investigation on the molecular basis for resistance to MLN, interactions between MCMV and SCMV for deployment of resistant varieties especially in areas where subsistent farmers' depend on continuous maize crop for food.

## CONCLUSIONS

Maize is a key cereal crop in Kenya and it associated with food security not only in Kenya but also in Sub-Saharan Africa. The average maize yield in Kenya is far below the global average because of biotic and abiotic stresses which puts a strain on production.

Transboundary diseases and pests are significantly increasing putting a strain on food and income security of millions of small-holder farmers.

In as much as these new diseases have previously been reported elsewhere, research is needed in Kenya so that the extent of the economic losses can be clearly defined and also new strategies to control and prevent these disease which are adaptable to the small holder setting in Kenya can be adapted.

With the lifting of the ban on use and cultivation of GMO in Kenya on October 2022, it is expected that researchers will use this technology to fast-track the release of improved maize varieties resistant to pests and diseases.

### Future perspectives

Maize is an important crop not only in Kenya but Sub-Saharan Africa at large. Therefore, the effect of biotic stresses has become a burden to especially the small-scale farmers. Hence, sustainable means of dealing with these are important for a food secure country. First, the available new technological advances and breeding methods should be applied to provide farmers with plant host resistance breeds and environmentally friendly control measures. Also, in most instances, these biotic stresses are detected too late. Improved technologies should therefore be used to develop early detection methods.

There is also a need to build a regional collective response to invasive pests and trans-boundary diseases.

Finally, there is a need to sensitize society on alternative crops to maize which is more tolerant to the pest and have superior nutritive value such as sorghum and millet.

## ACKNOWLEDGEMENTS

We would wish to acknowledge Dr. Erik Ohlson of Agricultural Research Service, US Department of Agriculture, corn, soybean and wheat quality research, Wooster, Ohio for reading the review article and giving his inputs and corrections and Dr. Alexander Mzula of Sokoine University of Agriculture (SUA) for his inputs and review of the article. The findings and conclusions of this study are those of the authors and do not necessarily represent the views of the funders.

### Funding

This study was funded by the Partnership for Skills in Applied Sciences, Engineering and Technology (PASET) through the Regional Scholarship and Innovation Fund (RSIF) awarded to Faith Njeru to carry out doctoral studies at SACIDS Africa Centre of Excellence for Infectious Diseases, SACIDS Foundation for One Health, Sokoine University of Agriculture, Morogoro, Tanzania. The L'Oreal-UNESCO For Women in Science Sub Saharan Africa program provided support for the APC. The funders had no role in study design, data collection and analysis, decision to publish, or preparation of the manuscript.

## Grant Disclosures

The following grant information was disclosed by the authors:

Partnership for Skills in Applied Sciences, Engineering and Technology (PASET).

SACIDS Africa Centre of Excellence for Infectious Diseases.

SACIDS Foundation for One Health.

Sokoine University of Agriculture, Morogoro, Tanzania.

L'Oreal-UNESCO for Women in Science Sub Saharan Africa program.

## Competing Interests

The authors declare that they have no competing interests.

## Author Contributions

- Faith Njeru conceived and designed the experiments, analyzed the data, prepared figures and/or tables, and approved the final draft.
- Angeline Wambua conceived and designed the experiments, analyzed the data, prepared figures and/or tables, and approved the final draft.
- Edward Muge conceived and designed the experiments, authored or reviewed drafts of the article, and approved the final draft.
- Geert Haesaert conceived and designed the experiments, authored or reviewed drafts of the article, and approved the final draft.
- Jan Gettemans conceived and designed the experiments, authored or reviewed drafts of the article, and approved the final draft.
- Gerald Misinzo conceived and designed the experiments, authored or reviewed drafts of the article, and approved the final draft.

## Data Availability

All data is available in the figures and text.

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
