# Peer review of "Major biotic stresses affecting maize production in Kenya and their implications for food security"

_PeerJ, doi:10.7717/peerj.15685_

## Round 0.1 · original submission · Major Revisions

Dear Dr. Njeru,

Your work has been reviewed by 3 independent experts. They all agreed that the paper could be published in PeerJ, but it had to be significantly revised beforehand. Please, read all the comments of the reviewers and respond to each of them.

With best regards,

Reviewer 1 ·

Basic reporting

Manuscript entitled „Major biotic stresses affecting maize production in Kenya and their implications for food security” represents the description of the serious problem that negatively affects corn yield in the region of interest to the authors. It appears that the authors have paid close attention to the effects on corn yield caused by biotic stress. The manuscript provides a comprehensive overview of the issues associated with the influence of various biotic factors on corn biology and corn yield. It is worth noting that the authors have presented the "algorithm" used in the literature search, which I consider to be very reliable and useful.
However, I have some serious comments about the form of the presentation:
- The title is confusing. The authors use the term "major biotic stresses" but do not provide any information on why these biotic stresses are most important for corn yield. Perhaps it is information about, for example, the percentage of yield damage?
- The abstract should be rewritten. It does not highlight the most important aspects of this work and does not encourage the reader to look more deeply into the article. In addition, the title refers only to biotic stresses, whereas the abstract also describes abiotic stresses.
- Maize (Zea mays L.) should be presented in this manner in the scientific literature. The summary is missing the "L.").
- I noticed the term "sub-Saharan Africa", but in the manuscript the authors use "Sub-Saharan Africa" - I do not know which form is correct, but please keep it consistent.

In the introduction section, although the FAO information is extensive, an economic approach is presented. I think this section should be shortened or the authors should focus on the effects of stress on plants.
The presentation of biotic stressors is inconsistent - for example, sometimes the authors present the "history of the disease" as a description of the biology of the disease (FAW). The subheadings should be standardized.

Experimental design

no comment

Validity of the findings

no comment

Additional comments

Some additional comments on many major and minor errors:

2: unnecessary repetition of FAO name
46: please specify what “major biotic stresses” actually means?
53: please keep the right order of literature
60: remove”;”, add”,”
74: the some remark as in the line 46
84: “attacks over 60 plants” – this is not a big number. You mean “over 60 plant species”?
90: keep the consequences in presenting % = 86 % or 86% in the manuscript
94: this is the remark from abstract
122: I think, form “However” should be the new sentence
124: look at the comment concerning the line 90
147: please change “5” on “five”
158: “WAN et al.” please rewrite whole reference section
201: Why “M” in marker-assisted selection term?
200/2002: The authors specify QTL as a plural form = loci, not locus, but in the nest sentence presented the results of study from 2020 as a QTLs (again plural). This is not correct
203: The sentence should be rewritten
218: The sentence should be rewritten. Please don’t start the sentence with “but”
225: remove Kansas as you don’t precise the region of Peru
227: potyviridae should be in italic?, et al., should not be in italic
238: However – this should be the beginning of new sentence
239: authors should use & or “and” when citing the references – keep the consequences
241: the same remark as the line 238
250: studies (only one study citied)
250-253: unnecessary repetition of Braidwood et al., 2018
266: Groote and his team (2013) – please keep the consequences in citing references
303: additional “(“
310: L.? this is the name?
320: I think the “,” is unnecessary
329: please keep in mind the plural form of QTL, change “5” on five
332: please keep the consequences when you are writing about QTL effects (large, big, main small)
334: change “7” on “seven”
385: Why “Chlorofluorocarbons” ? better “ chlorofluorocarbon”
390: again : his team”, please precise the right form
391: change the “4 and 5”on “four and five”
397: example of inconsistency in reference citing
400: again L. in italic, should be “L.”
400: “sp” should be not in italic
401: the same remark – spp
408: L. should not be in italic
410: and or &?
439: yield losses range from 0% - it is funny. 0 is not losses
441: again sub sahara
627-698: references, please provide form of presentation


The manuscript has many flaws in references description, it cannot be published without the right version of references presentation.
e.g. lines: 582, 590, 610, 621, and the here comes WAN (671 lines)
Figures:
Fig 1. Please combined the two photographs of (B) into one form
Fig 2. This is a scientific paper – do we need the photos of some resident of Kenya fighting with the insects? I doubt in it.

Reviewer 2 ·

Basic reporting

No comment

Experimental design

I found the logic flow/organization of this manuscript can be improved. For example, the paragraph of Lines 24-29 talks about abiotic and biotic stress, followed by a further discussion on the types of biotic stresses.This all makes sense but I don't understand what the purpose is of the next paragraph (Lines 37-42). In addition, Line 65 is back to abiotic stress again. I would recommend putting all the information about abiotic stress in one paragraph in the section of Introduction, and focus on various of biotic stresses in the main body of the paper.

Moreover, when talking about the types of biotic stresses, would it be helpful to group the pathogens? Like pests (FAM and Locusts), fungi (GLS and TLB), viruses (MLN), etc.

Validity of the findings

No comment

Additional comments

In Figure 1, panel B, the words in red are impossible to read. I would also suggest adding the figures showing symptoms from other pathogens.

·

Basic reporting

This manuscript aims to present five biotic stresses affecting maize production in Kenya and emphasizes the need to make deliberate efforts to control them. The manuscript was written in clear, unambiguous professional English. Field background information about maize production in Kenya and biotic stresses on maize duction was provided accurately and sufficiently. Published review literature on biotic stress in maize production is limited.

Minor suggestion on the figures and table:
1. The author included 2 figures that cover MLN and Desert Locusts. The author should add two more figures to cover the FAW and 2 foliar diseases (GLS and TLB).
2. Table 1 shows the average total maize production in Kenya from 2010 to 2018. As the author mentioned in the manuscript, abiotic stress also played an important role in the production of crops. In addition to year, area harvested (ha), and yield (t/ha), I would recommend adding information like average temperature, precipitation, whether experiencing drought, and the estimated loss if possible. This information could be important for summarizing Kenya's general maize production situation.

Experimental design

The manuscript summarized the literature about five biotic stresses that are affecting Africa. The manuscript introduced the history, the impact on maize production, the diagnosis, and the current control method for 5 biotic stresses including Fall Armyworm (FAW), Maize Lethal Necrosis (MLN) by Maize Chlorotic Mottle Virus (MCMV), Desert Locusts, Gary Leaf Spot (GLS), and Turcicum leaf Blight (TLB). The content is important and in the scope of the journal. The discussion of each biotic stress is well-organized and comprehensive.

However, the author combines GLS and TLB, two foliar diseases in one section. I would recommend minor adjustments from five to four biotic stress in the abstract.

Validity of the findings

Five different biotic stresses were discussed well. The manuscript clearly emphasizes the importance and difficulty of controlling those factors. The future perspectives were clear and useful.

Two suggestions I have for the author:
1. The author discussed fungicides, pesticides, and insecticides in the control sections. The information on the brand name and the mechanism for those treatments is missing. A brief introduction to the methods on apply pesticides or others will be helpful if possible.
2. The author did a great job covering most FAW control methods. However, besides what was discussed, the author did not cover the potential of microbial control for FAW. There have been some studies on microbial-based pesticides, for example:
https://doi.org/10.1016/j.jip.2019.01.006
https://doi.org/10.1111/jen.12565
https://doi.org/10.1079/PAVSNNR201914043
and a review: https://doi.org/10.1007/s10526-020-10031-0

---

## Round 0.2 · accepted · Accept

Dear Dr. Njeru,

Your work has been reviewed again by two independent experts. Both agreed that your work could be published in its current form. My congratulations!

With best regards!

Reviewer 1 ·

Basic reporting

The authors addressed all my comments and remarks.

Experimental design

The authors addressed all my comments and remarks.

Validity of the findings

The authors addressed all my comments and remarks.

·

Basic reporting

No Comment

Experimental design

No comment

Validity of the findings

No comment

Additional comments

No further comment